# Development of an In-Line Enzyme Reactor Integrated into a Capillary Electrophoresis System

**DOI:** 10.3390/molecules26195902

**Published:** 2021-09-29

**Authors:** Cynthia Nagy, Ruben Szabo, Attila Gaspar

**Affiliations:** Department of Inorganic and Analytical Chemistry, University of Debrecen, 4032 Debrecen, Hungary; nagy.cynthia@science.unideb.hu (C.N.); szabodavidruben@gmail.com (R.S.)

**Keywords:** immobilized enzyme reactor, adsorption, peptide mapping, capillary electrophoresis, human tear

## Abstract

The goal of this paper was to develop an in-line immobilized enzyme reactor (IMER) integrated into a capillary electrophoresis platform. In our research, we created the IMER by adsorbing trypsin onto the inner surface of a capillary in a short section. Enzyme immobilization was possible due to the electrostatic attraction between the oppositely charged fused silica capillary surface and trypsin. The reactor was formed by simply injecting and removing trypsin solution from the capillary inlet (~1–2 cms). We investigated the factors affecting the efficiency of the reactor. The main advantages of the proposed method are the fast, cheap, and easy formation of an IMER with in-line protein digestion capability. Human tear samples were used to test the efficiency of the digestion in the microreactor.

## 1. Introduction

The bottom-up approach used in proteomics is one of the most-often applied methods for the analysis of proteins. In this workflow, the sample is first mixed with a proteolytic enzyme (e.g., trypsin) to cut the long chains of proteins into smaller parts; then, the obtained peptides are separated by high-performance liquid chromatography (HPLC) or capillary electrophoresis (CE) before their detection with tandem mass spectrometry (MS/MS) [1]. The identification of peptides and proteins is facilitated by bioinformatic software and databases. In the standard protein digestion procedure, only a low concentration of the proteolytic enzyme is allowed in order to minimize the autolysis of the enzyme; therefore, the enzymatic reaction requires long incubation times (typically overnight) to complete the digestion [1,2]. Obviously, the necessity of such long incubation times is the bottleneck for fast, high-throughput, bottom-up proteomic analyses.

Enzymes can be immobilized on different supports in flow systems (immobilized enzyme reactors, IMERs). These reactors can also be formed in microfluidic devices (μIMERs), the sizes of which are smaller than a few cm^2^, allowing the use of sample/reagent volumes not exceeding the low microliter range. In the last few decades, a large number of articles have been published about such devices [3,4]. Owing to the high surface area-to-volume (S/V) ratio of microfluidic devices, enzymes can be immobilized in a high (surface) concentration. The attachment of enzymes to solid supports is most commonly done via adsorption [5,6], covalent bond [7], or bioaffinity linkage [8]. While most immobilization techniques described in the literature require a multi-step manipulation of the solid surface, the simplest and fastest option for immobilization is the adsorption of the enzyme on the wall of a microchannel or capillary. This approach takes advantage of the high propensity of large biomolecules like proteins to spontaneously adsorb onto solid surfaces. In our earlier works, we proved that trypsin can be directly adsorbed on the channel surface of a hydrophobic polydimethylsiloxane (PDMS) microchip, providing one of the simplest μIMER devices [9,10]. Digestion with this layer-bed-type immobilized enzyme reactor (empty channels) required less than a few minutes, and the enzyme remained active for a few hours (the regeneration of the reactor is easy and fast). While proteins are generally able to adsorb onto non-polar surfaces from solutions with a wide pH range (due to the large number of hydrophobic interactions), they may also strongly adsorb onto charged surfaces (via electrostatic interactions), provided that the net charge of the protein opposes that of the charged surface. It is a well-known fact that proteins adsorb on glass or fused silica surfaces, [11] but this phenomenon has been hardly utilized for IMERs [12]. Only a single paper [13] was found in which a trypsin-IMER was prepared by adsorption in a fused silica capillary for food analysis, but the efficiency of the immobilization or the digestion was not studied.

The high S/V of the solid support is a very important feature, which can be achieved in several ways, e.g., the integration of microparticles [4], monoliths [14], membranes [5], collocated monolith support structures (COMOSS) [15], or pillar arrays [16,17]. In addition, a high S/V ratio can be ensured in empty channel geometries, as well, if a long but small inner diameter (<75 μm) channel is applied [10].

The majority of published IMERs were applied in off-line mode, i.e., the digestion step is followed by a discontinuous analysis (separation and detection). The on-line and in-line analyses allow continuous processes. For the in-line analysis, the IMER is placed directly in the measurement process; there is no interface section between the reactor and the analytical device. In-line analytical systems are rare due to difficulties in fabrication or incompatibility/problematic interfacing of analytical steps; however, their advantages, including the possibility for fast, automatable analyses with a minimal sample consumption, are obvious.

In our present research, we created a μIMER by adsorbing trypsin onto the inner surface of a CE fused silica capillary in a short section. The CE capillary provided an excellent support due to its high S/V ratio as well as the possibility to create an in-line type of microreactor for CE-MS. We investigated the factors affecting the efficiency of the reactor. The applicability and main advantages of the proposed method were also studied.

## 2. Results and Discussion

### 2.1. The Characterization of IMER Efficiency

One of the key advantages of microreactors is their inherently high specific surface area. The diameter of CE capillaries matches those of microfluidic channels typically used as enzyme reactors (10–100 µm), providing a suitable platform for enzyme digestion. Trypsin-modified fused silica capillaries have long been used as IMERS [7,8,18,19,20,21,22,23,24]; however, in these cases, the capillary itself was not designed to show dual functionality (digestion and separation). The integration of these two analytical procedures into one unit (fused silica capillary) requires the application of experimental conditions that are favorable for both processes. This also implies that a compromise has to be made, since the acidic background electrolytes (BGE) commonly utilized for bottom-up proteomic studies in a CE-MS setting (e.g., formic acid, acetic acid) inhibit tryptic activity (pH optimum: ~8). Therefore, the development of conditions that satisfy both proteolytic and electrophoretic requirements are of the utmost importance. Regarding true in-line approaches, several achievements have been published; however, in these cases, UV detection was carried out, which considerably widens the array of suitable BGEs [25,26,27]. In these works, the trypsin immobilization procedure was typically a multi-step process, requiring an overnight (if not longer) incubation of the capillary. The immobilization of trypsin onto the initial short section of the capillary was carried out via encapsulation [25], covalent linkage [26], or electrostatic interactions, utilizing a poly(diallyldimethylammonium chloride) (PDDA)-anchored capillary surface [27].

Contrary to the approaches listed above, in our study, trypsin was bound to the surface via direct adsorption. The microreactor was formed by electrostatically binding trypsin on the inner surface of the CE capillary towards the inlet end (first few cms). The procedure for trypsin immobilization exploited the well-known behaviour of silica when in contact with aqueous solutions. Above pH~3, the silanol groups are deprotonated (pKa = 4.9), rendering the capillary surface negatively charged. Trypsin, however, has a net positive charge below its isoelectric point (pI = 10.3). The electrostatic attraction between the opposite charges is responsible for the immobilization. First, the presence of trypsin in the capillary was investigated. To verify the immobilization of trypsin, 1 mM BAEE was used as a substrate for the flow-through digestion CE analyses. Two sets of experiments were designed with varying sample introduction methods, the most important consequence of which was the varying contact times between the substrate and the surface-bound trypsin (Figure 1). In the first set (Figure 1A,B), three cases were examined: (1) the sample plug was introduced and immediately shifted through and beyond the IMER section by injecting BGE after the sample; (2) same as the previous, but the sample plug was allowed to stand for 1 min in the capillary after being shifted beyond the IMER, and (3) following injection, the sample plug was pushed to the middle of the IMER segment, where it was parked for 1 min, and it was finally shifted beyond the IMER. After injection, voltage was applied to separate the cleavage products, and peak areas were used to calculate the conversion%. An illustration of the three injection styles can be found in Figure 1A, and the corresponding efficiencies (expressed as substrate conversion%) are indicated in Figure 1B (Scenario 1), with a contact time of ~30 s yielding a substrate conversion of ~30%. This can be considered a reference point for the next two scenarios. At this stage, the stability of the trypsin layer was questionable; therefore, in order to make sure no considerable trypsin leaching occurred (which would promote in-solution digestion further along in the capillary), scenario 2 was investigated. Here, as mentioned before, the sample plug was paused after the IMER (distance between sample plug and IMER: ~0.25 cm) for 1 min. In the case of trypsin mobilization, this 1 min waiting time would have boosted the conversion ratio significantly. Since no obvious difference could be observed between the two cases regarding substrate conversion% values (Figure 1B), it was concluded that the generated trypsin layer could be considered stable. To improve efficiency and to prove that trypsin was, in fact, immobilized on the targeted section of the capillary, contact time was increased to 100 s in scenario 3 by pausing the movement of the sample plug for 1 min but, this time, within the IMER section. Since the laminar flow profile characteristics in such capillaries diminish considerable zone dispersion, analyte molecules are restricted to their original zone. This 1-min-long waiting step afforded abundant time for substrate diffusion, whereby the substrate could freely “travel” to the trypsin-coated surface. The efficiency of substrate conversion nearly tripled, as can be seen in Figure 1B. This striking improvement can be attributed basically to the longer contact time, which generally increases the occurrence of enzyme-substrate interactions. Since the substrate conversion ratios were in accordance with the assumed contact times in all three scenarios, we concluded that the IMER was, indeed, established in the given section. The sample injection scheme in the second set of experiments (Figure 1C–E) aimed at exploring how sample movement affects digestion efficiency. Experiment 4 is the simplest case, using the sample introduction method described previously in scenario 1, with the exception that here the sample plug was shifted adjacent to the IMER section, leaving no gap between the two zones (since the gap had no significance). Compared to case 5, where a lower pressure was applied for transporting the sample plug, a considerable change could be seen in the results (substrate conversion nearly doubled due to the slower motion of the sample and longer contact time (Figure 1D)). Next, we investigated the effect of intermittent sample progression in experiment 6), where the (50 mbar·8 s) injection used in scenario 4 was carried out in steps of four (50 mbar·2 s × 4). Finally, in case 7, the sample plug was parked in the middle of the IMER zone for 1 min. 

Judging by the results depicted in Figure 1D, the inclusion of a 1 min waiting time undoubtedly had the most favorable impact. However, this representation of data can be misleading if the effect of sample movement is to be evaluated, since it disregards contact time. Obviously, increasing the contact time yields better results; thus, one could rightfully argue that this was the underlying cause of improved efficiency. The normalized data in Figure 1E, however, demonstrate the utility of the slow, but continuous sample progression. The advantage stems from the fact that in this case the reaction was not limited by the diffusion of analytes to unoccupied enzyme molecules, because there was an uninterrupted flow, which provided constant contact with “new” surfaces. Sample injections are detailed in a tabular format in the Appendix A. To confirm sample behaviour and to investigate the radial motion of the transported analytes, COMSOL simulations were carried out (Appendix A). The mixing phenomenon was studied by introducing water and albumin solutions into the channel as two parallel liquid streams (1:1 ratio). The simulation is a good illustration of how mixing efficiency changes with decreasing linear flow velocities: below 3 mm/s, mixing was completed within 2 cm in a 50 µm ID capillary.

Factors affecting the formation and performance of the IMER were also explored using the BAEE substrate. Naturally, the pH has a huge impact on proteolytic efficiency. To explore pH-dependency, substrate conversion was studied in a pH range of 5–9 (Figure 2A). It is important to note that in a given experiment, the pH of the BGE, trypsin, and BAEE solutions were set to be identical. The results indicate an optimal pH value of 8, with a sudden decrease in IMER efficiency at pH = 9. Since the pH optimum of trypsin is ~8–9, the cause of this drastic decline was to be found in the lack of success concerning trypsin immobilization. With increasing pH, trypsin gradually loses its net positive charge; therefore, we assumed that at pH = 9, the small number of positively charged groups responsible for adsorption was insufficient, leading to an unsatisfactory surface concentration. Thus, in further experiments, the pH was chosen to be 8.

The concentration of the trypsin solution used for immobilization was investigated, as well. According to Figure 2B, increasing the trypsin concentration over ~10 mg/mL had no perceptible effect on IMER activity; at this point, the surface became saturated with the enzyme. Nevertheless, for further measurements, we used a 20 mg/mL trypsin concentration for immobilization.

Since trypsin is capable of autoproteolysis, it is important to suppress this activity during the immobilization procedure. Generally, tryptic activity can be quenched by acidification; therefore, we studied the effect of lowering the pH of the trypsin solution (pH range: 2–8). Figure 2C suggests a substantial decrease in enzyme activity at pH < 4, which is presumably caused by the inadequate immobilization. At such low pH values, the capillary surface loses its abundance in deprotonated silanol groups, providing reduced potentiality for trypsin attachment. The lowest pH value at which the diagram shows no sign of deterioration in IMER performance is pH = 4; therefore, this value was used for subsequent analyses.

It was also essential to investigate the stability of the immobilized trypsin layer without its regeneration between consecutive runs. Figure 2D clearly proves the necessity of regeneration after each individual measurement. There can be several factors contributing to the reduced activity of the IMER. The application of voltage or simply the capillary flushing procedure might mobilize enzyme molecules from the surface, resulting in a looser, less compact layer of trypsin. It is also possible that due to the interaction between the capillary surface and certain amino acid side chains, the tertiary structure of surface-bound trypsin changes with time, which leads to a loss in function. More details pertaining to this drop in IMER performance are given in Section 2.2.

All measurements described above were carried out in triplicate, according to Appendix A, where the trypsin immobilization procedure was performed 3× to ensure the formation of a uniform, tightly arranged trypsin layer. The importance of repeating trypsin administration was demonstrated in Appendix A, where IMER activity is shown as a function of the number of repetitions. The points on the graph suggest a tendency similar to that of a saturation curve, were saturation is fulfilled by the third repetition. 

### 2.2. Digestion of HSA Using CE-MS 

The developed flow-through microreactor showed encouraging results with the BAEE substrate; thus, the next step was to assess its capability to digest proteins in a CE-MS workflow. Because proteins show a much higher level of complexity than BAEE, it was uncertain whether the previously utilized injection program with a ~100 s contact time would yield adequate digestion performance. Therefore, a series of in-line digestions were performed with varying residence times (0.3–7.5 min), using human serum albumin (HSA) as model protein (Figure 3). The 59% sequence coverage (SC) value obtained using the shortest (0.3 min) contact time (Figure 3A) was surprisingly high at first glance, considering the scarcity of peaks on the base peak electropherogram (BPC). The shape of the peaks implies the existence of large peptides. Indeed, most of the identified peptides contained two uncleaved sites. Naturally, the gradual increase in contact time resulted in a higher number of peaks appearing on the electropherograms, suggesting improved digestion efficiency. Therefore, subsequent experiments were conducted using a 7.5-min-long residence time. To verify the reliability of the developed IMER, five consecutive in-line digestions were carried out with the HSA (Figure 4). The five electropherograms showed outstanding conformity upon a visual inspection, proving the proper repeatability of in-line digestions. RSD% data were calculated for three randomly selected peaks in the five electropherograms, which were 0.75, 1.08, and 1.02% for migration times and 37.9, 39.8, and 37.4% for peak areas. The repeatability of the CE measurements was also examined, with five consecutive CE runs of the same in-solution HSA digest (Appendix A). The calculated RSD% values of three randomly chosen peaks were 1.21, 0.89, and 0.97% for migration times and 11.2, 16.3, and 21.2% for peak areas. Proteolytic efficiency was also investigated by comparison with the traditional in-solution digestion (Figure 5). The peak patterns of the mirrored electropherograms were in good agreement with each other. For a more in-depth evaluation, SC% values and the ratio of missed cleavage peptides (%) were calculated for both cases (Figure 6). Summed area-under-the curve (AUC) values of the peptides containing uncleaved sites were compared to the summed AUC values of all peptides present in the electropherogram. While SC% values showed no significant difference (60–70%), in-line digestions seemed to have generated peptides containing one missed cleavage (rarely two) in a higher abundance than with in-solution digestions. These missed cleavages are indicated in Appendix A, where the peaks are denoted with the corresponding peptides, highlighting those R (arginine) and K (lysine) residues where hydrolysis did not occur. The existence of uncleaved bonds in the case of in-line digestions actually allowed the identification of short sequences that were not detected for the in-solution digests. These sequences typically contain two or three amino acid residues (e.g., peptide YK at position 185–186, peptide FK at 35–36, or CCK at 582–584), which were not detectable with the MS method used. Despite the 50–2200 *m*/*z* mass range, sensitivity drastically decreased at *m*/*z* < ~400, inhibiting the detection of such short peptides. Since Byonic searches were conducted against the Swissprot database, it was possible to monitor the autolytic behaviour of trypsin, as well. Naturally, in-solution digested HSA peptide mixtures contained autolysis-derived tryptic peptides; however, the analyses using the in-line IMER also yielded these peptide identifications. The presence of autolytic peptides can be attributed to the mobilization of trypsin from the surface. As the desorbed trypsin molecules move further along the IMER section, they can either be digested by the immobilized enzyme, or the surface-bound trypsin is hydrolyzed by the detached trypsin. Either way, trypsin mobilization (among other factors mentioned in Section 2.1) results in a deterioration of IMER performance, as demonstrated in Figure 2D.

### 2.3. Application of the In-Line Microreactor for CE-MS

The ultimate test of IMER reliability and efficiency is its capability of handling complex protein samples. For this reason, human tear samples were digested in-solution and in-line. The measurements yielded comparable results to each other, as well as to our previous study utilizing a microchip-IMER (off-line approach) for tear proteome digestions [28]. Sample pretreatments were identical. The migration time and peak area RSD% values were determined for five consecutive in-line CE-MS measurements. The calculated RSD% values based on three randomly chosen peaks were 1.75, 1.28, and 1.09% for migration times and 23.6, 29.7, and 46.1% for peak areas. In Figure 7, only those proteins that were identified by at least two unique peptides are listed. The sequence coverage values of these 11 proteins are depicted for both in-solution and in-line workflows. The displayed SC% values are the average of five consecutive digestions. Overall, in-line digestions provided higher SC% values, most probably because of the reasons described in Section 2.2. Taking a closer look at the confident protein identifications, in-line digestion typically produced a higher number of sequence hits and unique peptides. This phenomenon can be attributed to the presence of uncleaved sites, generating a more complex set of peptides in the mixture. A given peptide containing uncleaved bonds can also be present in its fully digested form, thereby leading to a higher number of sequence hits. Despite the missed cleavages, the in-line IMER did not fall short of protein hits compared to the in-solution digestion. However, it is possible that in addition to yielding a higher number of confident protein identifications, using nanoESI would reveal the subtle differences between the efficiencies because of its improved sensitivity. Sensitivity can also be enhanced by exploiting on-line preconcentration strategies. This is substantiated by the fact that preliminary experiments utilizing the stacking technique (with 1 M formic acid as BGE and larger sample loading: 50 mbar·120 s) offered a higher number (nearly double) of protein identifications (data not shown). Stacking was only marginally investigated with in-solution tear digests; however, the application of acidic BGEs for in-line digestion would require a whole new set of method optimization, which is outside the scope of the present manuscript. Nevertheless, the developed in-line µIMER proved to be a promising tool for proteomic studies.

## 3. Materials and Methods

### 3.1. Reagents and Solutions

Analytical grade reagents were used. Porcine pancreas trypsin (Type IX-S, lyophilized powder, Sigma, St. Louis, MO, USA) was used as a proteolytic enzyme. For in-solution digestions, trypsin was freshly prepared in double-deionized water; for IMER digestions, the enzyme was dissolved in acetic acid (pH = 4) solution. *N*-α-benzoyl-l-arginine ethyl ester hydrochloride (BAEE) substrate was used to examine the parameters affecting IMER efficiency. Human serum albumin (HSA) (Sigma) and human tears were digested to test the reliability of the in-line IMER in a CZE-MS/MS setup. BAEE, urea, dithiothreitol (DTT), iodoacetamide (IAM), NH_4_HCO_3_, and NH_4_Ac stock solutions (all Sigma products) were prepared in double-deionized water (Elix-3, Millipore, Darmstadt, Germany). HCl, NaOH, ammonium hydroxide, acetic acid (HAc) solutions, isopropanol, methanol, and acetonitrile were purchased from VWR (Radnor, PA, USA).

### 3.2. Instrumentation, Software

Measurements relating to the investigation of µIMER behaviour with a BAEE substrate were carried out with HP ^3D^CE instruments (Agilent, Waldbronn, Germany), using on-capillary DAD detection. The electropherograms were recorded and processed by ChemStation software (ver.: 7.01, Agilent, Santa Clara, CA, USA, 2011). The analysis of digested protein samples was performed with a 7100 model CE instrument (Agilent) coupled to a maXis II UHR ESI-QTOF MS (Bruker, Bremen, Germany) via a CE-ESI Sprayer interface (G1607B, Agilent, Santa Clara, CA, USA). Sheath liquid was delivered with a 1260 Infinity II isocratic pump (Agilent, Santa Clara, CA, USA). The CE instrument and the pump were controlled by OpenLAB CDS Chemstation software (A.02.17, Agilent, Santa Clara, CA, USA, 2017). The MS was operated by otofControl version 4.1 (build: 3.5, Bruker, Bremen, Germany 2017); the obtained peptide maps and mass spectra were processed by Compass DataAnalysis version 4.4 (build: 200.55.2969, Bruker, Bremen, Germany, 2016). The generated peak lists were exported in MGF format. Byonic software (ver.: 3.9.6., Protein Metrics, Cupertino, CA, USA, 2020) was used for peptide/protein identification. The datasets were searched against the Swissprot database. Byonic runs were performed with the following settings: fully specific digestion; missed cleavage tolerance: 2; precursor mass tolerance: 15 ppm; fragment mass tolerance: 40 ppm; carbamidomethylation (+57.021464 Da) at Cys as a fixed modification; deamidation (+0.984016 Da) at Asn and Gln as variable modifications.

Flow simulation was carried out with COMSOL Multiphysics (ver.: 5.3a, Burlington, MA, USA, 2017), which is a finite element (FEM) analysis-based simulation software. The mesh size was set to extremely fine.

### 3.3. Preparation and Operation of the Microreactor in the CE Capillary

For decent immobilization, the capillary needed to be preconditioned with the BGE (NH_4_Ac, pH = 8), which imparted a uniform, negatively charged surface to the capillary, ensuring proper attachment of the positively charged trypsin molecules. The goal was to create the microreactor on a relatively short section of the capillary, simply by introducing the trypsin solution (20 mg/mL). A 50 mbar·8 s injection of the trypsin solution corresponded to a 1.04 cm capillary length. Setting a 1-min-long waiting time allowed trypsin molecules to diffuse to the surface. Subsequently, trypsin was washed out from the capillary with the BGE, applying −50 mbar·12 s (using a 1.5× larger volume). This 3-step procedure was repeated to ensure the formation of a stable, uniform trypsin layer. BGE was then flushed (−50 mbar·120 s) through the capillary into a vial containing deionized water, which guaranteed the elimination of unbound trypsin from the conduit (to inhibit accidental in-solution tryptic activity). As a “finishing touch”, the capillary was conditioned with fresh BGE solution (1 bar·60 s).

For in-line digestions, the injected sample plug (50 mbar·2 s) was slowly transported through the IMER section (L = 1.04 cm) by introducing the inlet BGE (50 mbar·8 s). Finally, upon application of voltage, the separation of peptides commenced. 

The Hagen–Poiseuille formula was used to calculate the injection scheme. A detailed tabular summary of the preconditioning and injection program can be found in the Appendix A.

### 3.4. Operation of the μIMER-CE-MS System

The μIMER-CZE-MS/MS platform was used for the digestion of protein samples (HSA, human tear). The preconditioning and injection scheme had to be slightly modified for two reasons: (i) a ~2.6× longer capillary was used for CE-MS, which affects the injected amount, and (ii) the sample matrix and composition are more complex in the case of peptide digests. Trypsin was dissolved in acetic acid (pH = 4) to acquire a concentration of 20 mg/mL. Since we had a longer separation capillary, the length of the IMER section was nearly doubled (L = 1.95 cm) to provide an increased path length for digestion to occur. Therefore, following adequate capillary conditioning with the BGE, the trypsin solution was introduced using 50 mbar·40 s. After a 1-min-long waiting time, the solution was withdrawn (−50 mbar·60 s). This 3-step process was repeated two more times. Sample injection steps were also optimized to better accommodate the challenges arising from sample complexity. The injected sample plug (50 mbar·6s) was slowly pushed to the middle of the IMER segment, using the lowest possible pressure the CE instrument is able to exert reproducibly (15 mbar·60 s). The subsequent step included a 5-min-long waiting period. Finally, the sample was pushed further to the end of the IMER segment (15 mbar·60 s), after which the separation was immediately started. The described injection program was utilized for protein samples, if not stated otherwise. A tabular representation of the optimized preconditioning and injection parameters can be found in the Appendix A.

### 3.5. CZE Separations

For CZE-UV analyses, polyimide-coated fused silica capillaries of 34 cm × 50 µm id. (Polymicro, Phoenix, AZ, USA) were used (L_eff_ = 26 cm). The BGE consisted of 20 mM NH_4_Ac (pH = 8); the applied voltage was 25 kV. Sample introduction was carried out at the anodic end of the capillary. The formation of the immobilized trypsin layer was achieved in a fully automated manner, integrated into the preconditioning step as described previously in Section 3.3. Detection was carried out at λ = 230 nm (bandwidth: 4 nm; response time: 0.3 s). The capillary was post-conditioned with 1 M NaOH (1 bar, 5 min). 0.1 M HCl (1 bar, 1 min) and BGE (1 bar, 30 min). The surface-bound trypsin layer was regenerated prior to each measurement.

For CZE-MS/MS analyses, 90 cm x 50 µm id. fused silica capillaries were used. The BGE was 40 mM NH_4_Ac (pH = 8); the applied voltage was 22 kV. The generation of the trypsin layer and the sample introduction were conducted as detailed in Section 3.4. The post-conditioning step involved washing with acetonitrile (4 bar, 2 min), water (4.5 bar, 2 min), and BGE (4.5 bar, 2 min). Sheath liquid consisted of isopropanol:water = 1:1 + 0.5% HAc and was delivered at a flow rate of 6 µL/min to establish electric connection and stable electrospray formation. The following parameters were applied for MS acquisition: positive polarity mode; nebulizer pressure: 0.5 bar; dry gas temperature: 180 °C; dry gas flow rate: 4 L/min; capillary voltage: 4500 V; end plate offset: 500 V; MS spectra rate: 3 Hz; MS/MS spectra rate: 1–4 Hz; mass range: 50–2200 *m*/*z*. Fragment ions were generated by collision-induced dissociation (CID). Na-acetate adducts enabled internal m/z calibration.

### 3.6. Enzymatic Digestion of Protein Samples

To test the proteolytic efficiency of the developed microreactor, a model protein (HSA) and tear samples were digested. HSA samples were pretreated as follows: ~4 mg protein was dissolved in 100 μL 25 mM NH_4_HCO_3_, and immediately, 300 μL 8 M urea was added to induce denaturation (30 min, room temperature (RT)). A total of 40 μL 100 mM DTT was pipetted into the solution to reduce the disulfide bridges (1 h, 37 °C); then, 40 μL 200 mM IAM was added to alkylate the sulfhydryl groups of the cysteines (45 min, RT, dark). The mixture was diluted with 2 mL 25 mM NH_4_HCO_3_ in order to decrease the urea concentration below 1 M. 

Non-stimulated human tears were collected with a sterile capillary tube [29]. Tear samples were prepared as follows: 0.9 mg urea, 0.3 μL 100 mM DTT, 0.3 μL 200 mM IAM, and 9.5 μL 25 mM NH_4_HCO_3_ were added to 5 μL tear (the proper incubation times and temperature values were described above). The pretreated samples were stored at −20 °C until digestion.

For control samples, digestions were performed in-solution prior to the application of the μ-IMER. For these control digestions, 80 µL and 0.3 µL freshly prepared 1 mg/mL trypsin solution was pipetted into the HSA and tear sample mixtures, respectively. After overnight incubation (16 h, 37 °C), the reaction was terminated with the addition of 1% HAc to a 0.1% HAc final concentration.

The conditions for μ-IMER digestions are discussed in detail in Section 3.3 and Section 3.4.

## 4. Conclusions

In our study an in-line IMER was developed in a CE-MS platform for performing bottom-up proteomic analysis. The enzyme was immobilized at the entrance of the CE capillary (1–2 cm) via adsorption, exploiting the opposing charge polarities of trypsin (net positively charged below its pI (=10.3)) and the capillary wall (deprotonated above pH: ~3). The optimization of trypsin immobilization and IMER performance were investigated using a BAEE substrate. The developed method was utilized for the in-line IMER-CZE-MS/MS analysis of HSA and human tear samples, yielding satisfactory results. In-solution digests of the same protein samples produced comparable results, proving the applicability of the designed in-line system. The proposed immobilization of trypsin is a straightforward and fast procedure (~few min), requiring no special chemicals or the multi-step pretreatment of the capillary surface. The in-line nature of the IMER circumvents the manual handling of samples between each processing step, which often leads to the disruption of sample integrity or sample loss. The developed IMER is also unique in the sense that in addition to the digestion and subsequent separation steps being performed in a single capillary, the immobilization step can also be carried out without manual manipulation. Thus, immobilization, digestion, and separation are performed in a fully automated manner, providing one of the simplest in-line proteomic workflows. Furthermore, the platform enables us to conduct remarkably economical analyses, since due to the small dimensions of the IMER, only ~nL volumes of trypsin and protein samples are required.

## Figures and Tables

**Figure 1 molecules-26-05902-f001:**
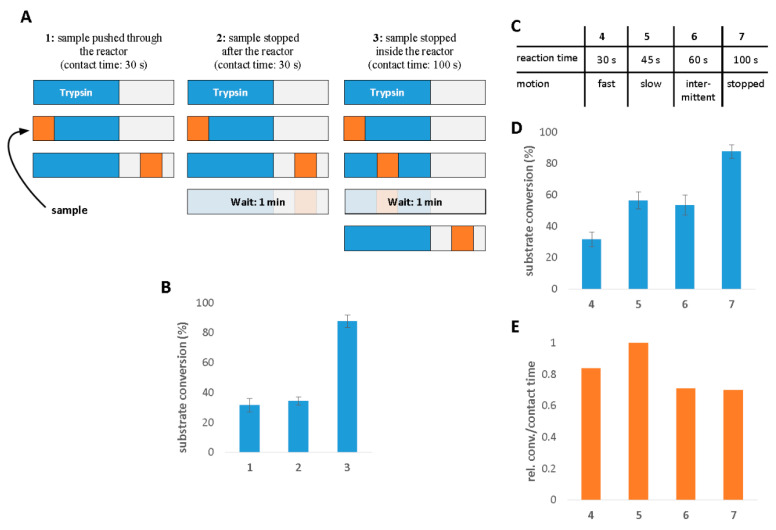
Experiments to verify the immobilization of trypsin: (**A**) illustrations and descriptions of the experiments 1–3; (**B**) the effect of sample movement (1–3) on the enzyme reaction; (**C**) descriptions of the sample movement conditions 4–7; the effect of sample movement (4–7) on the enzyme reaction (**D**). Substrate conversions divided by the contact time and normalized to 1 for sample movements 4–7 (**E**). Measurement parameters: fused silica capillary, ID = 50 μm, 34 cm length (Leff = 26 cm); BGE: 25 mM NH_4_Ac pH = 7; trypsin solution: 20 mg/mL; substrate: 1 mM Nα-Benzoyl-L-arginine ethyl ester; voltage: 25 kV; UV detection: 230 nm; sample injection: 50 mbar·2 s. Preconditioning and injection parameters are detailed in Appendix A and Appendix A, respectively.

**Figure 2 molecules-26-05902-f002:**
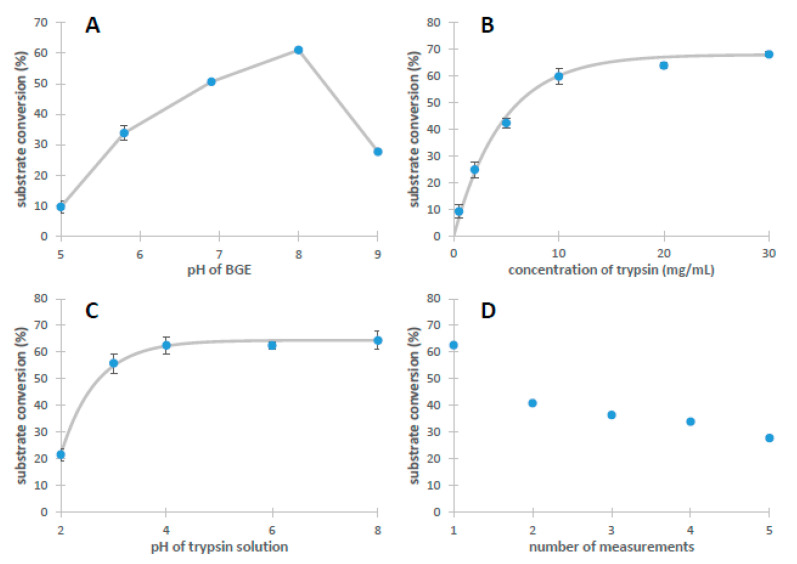
Parameters such as pH of the BGE (**A**), concentration of trypsin solution (**B**), pH of trypsin solution (**C**), and number of the measurements (**D**) affecting the activity of the reactor. Conditions were the same as in Figure 1. Preconditioning and injection parameters are detailed in Appendix A.

**Figure 3 molecules-26-05902-f003:**
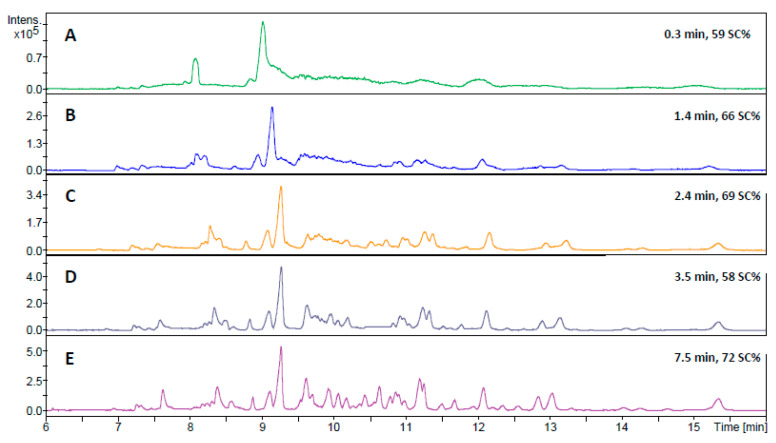
CZE-MS measurements after in-line digestion of human serum albumin with increasing contact times (0.3 min–7.5 min). The approximate contact times and the sequence coverage% values obtained are shown on the right. Conditions were the same as in Figure 1. But BGE: 40 mM NH_4_Ac, pH = 8; trypsin solution: 20 mg/mL, pH = 4; fused silica capillary: 90 cm length; injection parameters were 50 mbar for 6 s (**A**); sample: 50 mbar for 6 s, then BGE: 15 mbar for 60 s (**B**); sample: 50 mbar for 6 s, then BGE: 120 s for 15 mbar (**C**); sample: 50 mbar for 6 s, BGE: 15 mbar for 60 s, 60 s waiting, and then 15 mbar for 60 s (**D**); sample: 50 mbar for 6 s, then BGE: 15 mbar for 60 s, 300 s waiting, and then 15 mbar for 60 s (**E**). MS: positive ionization mode; spectra rate: 8 Hz; MS/MS frequency: 1–4 Hz; nebulizer pressure: 0.5 bar; dry gas temperature: 180 °C; sheath-liquid: IPA: water = 1:1 + 0.1% HAc.

**Figure 4 molecules-26-05902-f004:**
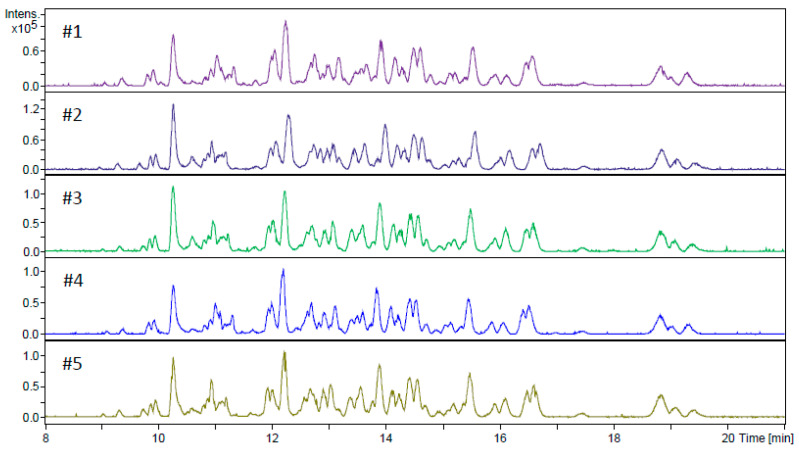
5 repetitions of CZE-MS measurements after in-line digestion of human serum albumin. Conditions were the same as in Figure 3; contact time: 7.5 min. Preconditioning and injection parameters are in Appendix A.

**Figure 5 molecules-26-05902-f005:**
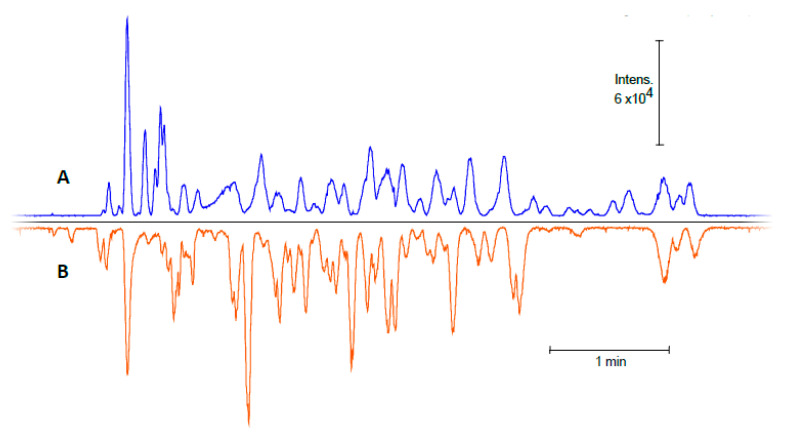
Comparison of the obtained CZE-MS electropherograms after in-solution (**A**) and in-line digestion (**B**) of the human serum albumin sample. Conditions for in-line digestion are as described at Figure 4; for the measurement of in-solution digest, the formation of the trypsin layer was omitted.

**Figure 6 molecules-26-05902-f006:**
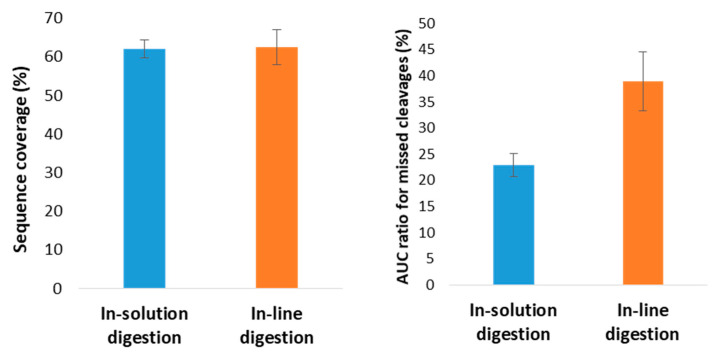
Comparison between in-solution and in-line digestion of human serum albumin in terms of sequence coverage% (**A**) and the ratio of missed cleavage peptides (**B**). The values shown are the averages of 10 consecutive measurements, using the parameters described in Figure 4.

**Figure 7 molecules-26-05902-f007:**
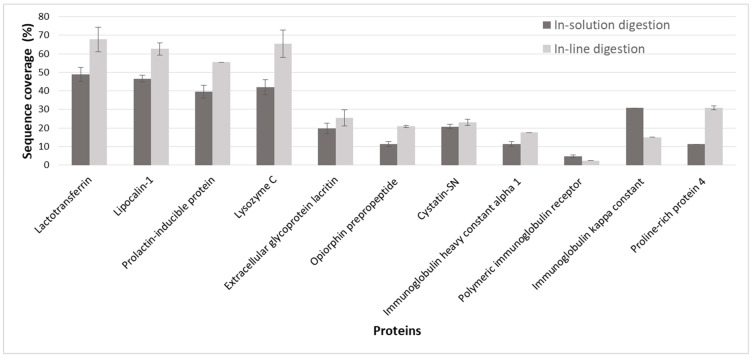
Identified proteins and their sequence coverage values in human tear sample with in-solution and in-line digestion using CZE-MS/MS. A total of 11 proteins identified with at least two unique peptides are shown. Preconditioning: BGE (1 bar·180 s), trypsin solution ((50 mbar·100 s, 60 s wait time, −50 mbar·120 s) × 3), water (−50 mbar·120 s), and BGE (1 bar·60 s). Injection: sample (50 mbar·60 s), BGE (15 mbar·70 s), 300 s wait time, and BGE (15 mbar·70 s). Other parameters were the same as in Figure 4.

## Data Availability

Not applicable.

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
