# Peer review of "Development of an In-Line Enzyme Reactor Integrated into a Capillary Electrophoresis System"

_molecules, 2021, doi:10.3390/molecules26195902_

Round 1

Reviewer 1 Report

The manuscript entitled "Development of an in-line enzyme reactor integrated into capillary electrophoresis system" written by Cynthia Nagy, Ruben Szabo and Attila Gaspar can be published as it stands.

Author Response

We thank the Referee 1 for her/his good opinion about our manuscript.

Reviewer 2 Report

Directly adsorption of enzym in silica capillary for digestion is  very interested in the connection with electrophoresis separation. Since the limitation of capacity and the digestion distribution of the samples, it remained as potential useful for long time. The manuscript described a very positive results and can be accepted for publication.  The detail comments are listed as below. 

  1. Since the amount of enzyme adsorbed in the end of capillary are critical for digestion, did the authors measure the surface concentration of enzyme or absolute amount of enzyme in the end?
  2. The kinetics of digestion is important for this immobilzation protocol, the results for digestion in different times should be presented. 

Reviewer 3 Report

In the present manuscript Nagy C., et al. report on the development and some performance characteristics of a very simple and economically reasonable in-line enzyme reactor for capillary electrophoresis-based protein analysis by mass-spectrometry. The manuscript is very well written, and scientifically sound. Although -as was also pointed out by the references discussed by authors in the introduction section- lacking immediate conceptual novelty, I endorse publication of this manuscript with a few minor additions.

major points:

- From the data provided it remains unclear how much trypsin was actually immobilized on the surface of the in-line reactor-section of the capillaries, and -most importantly- if the conditions used in the respective control experiments (i.e. protein to proteinase ratio used for in-solution digests) are appropriate. I do acknowledge that the determination of the exact amount of trypsin immobilized on the surface of the reactors may be technically very challenging, but still recommend the authors providing at least a rough estimate, and discussing the validity of the control experiments performed.

- "% sequence coverage" and "average of missed cleavages" are poor parameters to compare the degree of proteolytic degradation (i.e. in the case of HSA and tear samples). Instead I recommend comparing e.g. the summed area-under-the curve (AUC) value of all identified peptides, and the AUC values of selected missed-cleavage peptides.

- Throughout the manuscript, it is unclear if trypsin auto-lysis was considered at all. For example, it is unclear whether the amino-acid sequence of porcine trypsin was (e.g. as potential contaminant) included in the protein sequence database used for the automated peptide identification by mass-spectrometry. Please clarify, elaborate and discuss!

- Similarly, in fig 2.D (and lines 201-208), the authors report on and discuss the decreased performance/stability of the trypsin layer over consecutive runs, yet no mentioning of autolysis being a possible cause for this observation. Please clarify, elaborate and discuss!

Reviewer 4 Report

This manuscript describes the design and the optimization of an in-line enzyme reactor, coupling to a capillary electrophoresis system. Trypsin is first immobilized by electrostatic attraction between the protonated trypsin and the deprotonated fused silica surface. N-α-benzoyl-L-arginine ethyl ester hydrochloride (BAEE) is used as the substrate to optimize the performance of the reactor. Parameters such as the flow rate, reaction time, pH of the electrolyte and the trypsin buffer are optimized. The performance of the reactor is further evaluated by the in-line digestion of human serum albumin. The migration times show good reproducibility (RSD ~1%) and are comparable to the in-solution digestion. However, the peak area varies. The performance of the reactor is then further evaluated by the line digestion of human tear samples. Most proteins detected have a better sequence coverage in the in-line digestion than the in-solution digestion.

Minor comment: It would be ideal if the author can show the reproducibility of the tear proteome in the analysis.

The design of the in-line enzyme reactor is innovative and its performance is well characterized in the manuscript. It would be interesting to see the in-line reactor perform in PTM analysis, or if it is compatible with other enzymes such as PNGase F for glycoproteomic analysis.

Round 2

Reviewer 3 Report

The authors revised their manuscript according to my recommendations. I endorse publication of the paper in its present form.